# A sense of commitment to activity on Facebook: Evidence from a web-based paradigm

Chiara Brozzo[1], John Michael [2]*

**1** University of Barcelona, Barcelona, Spain, **2** University of Milan, Milan, Italy

* Johnmichael.cogsci@gmail.com

## Abstract

The present study was designed to test whether actions on Facebook such as commenting on others' posts can create a sense of commitment to continue performing similar actions in the future. Across four online experiments, we found evidence that regularly commenting on others' Facebook posts generates a sense of commitment to commenting on similar posts in the future, leading people to feel worse about not commenting on a post if they have done so regularly in the past than if they have not done so, and leading them to expect a Facebook friend to be more disappointed if they do not comment when there has been this previous history of commenting. These findings may contribute to illuminating the feelings associated with social media use, including its compulsive aspects and its effects on well-being.

## 1. Introduction

Suppose that today you open your Facebook feed, and you are notified that it is Sam's birthday, Sam being one of your many Facebook friends. You decide to wish her a happy birthday by posting on her Facebook wall. Before doing so, however, you notice that a number of other Facebook friends of yours also have birthdays coming up in the next few days. Do you now feel under pressure to wish them a happy birthday too, just in case they notice you have wished a happy birthday to Sam, and might feel disappointed if you do not also wish them a happy birthday? If so, do you feel under pressure to open your Facebook feed regularly over the next few days so as to be reminded to wish your other friends a happy birthday? Or do you end up reconsidering your seemingly harmless decision to wish Sam a happy birthday, so as to escape any pressure to wish a happy birthday to more of your Facebook friends in the future?

Our starting point in the current study was the conjecture that such experiences may be explicable as arising from a sense of commitment. Previous research has demonstrated that two agents A and B, by repeatedly performing actions, can establish a sense of commitment between them, even when no explicit agreement has been made among them [1–4]. For example, if Polly and Pam have habitually met on the balcony of their office building to smoke a cigarette together during their coffee break for the past three years, if one day Polly fails to show up, participants reliably judge that Pam will feel that Polly owes her an explanation for not showing up. This is so even though Polly and Pam never entered into any explicit agreement to meet up on the balcony to smoke a cigarette together during their coffee break [1, 2].

**Data Availability Statement:** As we state in the paper: The data and R code can be accessed here: https://osf.io/vu6h4/?view_only= 940e248296c44e9f92c2af357440b77c.

**Funding:** JM was awarded a Starting Grant (679092: Sense of Commitment) by the European Research Council: https://erc.europa.eu. CB's research was supported by the Starting Grant ReConAg 757698 from the European Research Council awarded to Joshua Shepherd under the Horizon 2020 Programme for Research and Innovation. The open access fees were kindly covered by the research vice-rectorate of the University of Barcelona and by the University of Milan. The funders played no role in the study design, data collection and analysis, decision to publish, or preparation of the manuscript.

**Competing interests:** The authors have declared that no competing interests exist.

Indeed, it has been shown that the strength of this sense of commitment can be modulated by many subtle situational factors, such as the number of times that interaction has been repeated in the past, and the amount of effort that the agents have invested in the interaction. To return to the Polly and Pam example, the longer Polly and Pam's meetings on the balcony repeat over time, the stronger their mutual sense of commitment will be. If their habitual meeting has been going on for only three days, rather than for three years, participants tend to agree to a significantly lesser extent that Pam will feel that Polly owes her an explanation when she does not show up [1, 2]. Moreover, if Pam has to walk up five flights of stairs to get to the balcony, participants tend to judge that Polly will feel a greater sense of commitment than she would if Pam only had to walk a few steps [1–4].

Thus, there is considerable evidence that face-to-face dyadic interactions create a sense of commitment even in the absence of explicit agreements. But can a sense of commitment also be engendered through remote, on-line interactions? The present study addressed this question by investigating whether behaviours on social media could give rise to a sense of commitment in a similar way that face-to-face dyadic interactions are known to do. The following testimony by Sharon Baldessari, 32 years old, working in technology sales in New York, hints towards a positive answer:

> "Right around the time of George Floyd's murder and when the black square went viral in solidarity with Black Lives Matter (BLM) [. . .] I felt enormous pressure to post it even though I didn't know what it meant. I felt like an invisible hand was compelling me to repost–otherwise, it would mean I didn't care about BLM and that I was a racist. I realized this was absurd, and I rejected being completely defined as a person by what I posted or didn't" [5].

To implement our study, we elected to focus on the context of Facebook, given that Facebook is one of the most widely used social media [6]. It is known that users sometimes engage in a problematic way with Facebook. One study [7], for example, reported a correlation between Internet addiction and so-called problematic Facebook use, and argue that the latter could be considered a subtype of the former. What contributes to a user's engagement with Facebook, especially one that may be regarded as addictive?

One possible explanation is that people overestimate the extent to which spending time on Facebook is going to make them feel better [8]. This explanation concerns Facebook use in general, whether one is passively consuming content or also actively posting items or reacting to others' posts. In this paper, we are interested in elucidating what contributes to a specific kind of Facebook use, which is that of reacting to other people's posts—for example, "liking" someone's post—rather than, say, just scrolling the news feed without reacting to any specific post. What drives the need to react to other people's posts?

Previous studies have identified "likes" as a kind of social currency. In line with *sociometer theory* [9], according to which self-esteem is a sociometer, that is, a measure of the extent to which one views oneself as a valued member of certain social groups, some studies have proposed that receiving "likes" to a picture that one posts can boost one's self-esteem, by giving them a sense of acceptance [10]—and, furthermore, when the posted picture depicts one's face, a high number of received "likes" will reduce face dissatisfaction [11]. This has the potential for explaining the motivation for engaging in Facebook activity that leads to posting items that are likely to receive "likes". Posting a certain kind of content on Facebook, that is, is motivated by the expectation to receive a certain kind of social currency. But what is the motivation behind *giving* social currency, in the form of "likes", and, more generally, behind reacting to others' posts?

One study [12] identified a possible response by investigating the neural correlates of liking images. Their studies revealed that liking images posted by other people activates areas of the brain that are associated with reward. Their hypothesis is that giving a "like" to another's picture is perceived as an "adaptive prosocial response, similar to giving to charity or providing social support" [12, p. 706]. However, the fact that their study looks specifically at the neural activity associated with liking pictures, as opposed to other kinds of Facebook posts, allows for the fact that this reward response could partly be due to the liked photographs having pleasant features that elicit positive feelings.

In sum, we have reason to believe that liking other people's posts might *feel good* in the way that other forms of giving social currency do. Of course, we must be cautious in interpreting and in generalising these results—in particular because the posts for which the liking activity was investigated were limited to pictures, whereas Facebook posts are not limited to pictures, but may also consist entirely of bodies of texts (similarly to Twitter, and unlike Instagram). Moreover, as previously mentioned, the pleasant feeling might be to some extent be determined by the characteristic of the picture to which one is giving a "like".

In the current study, we sought to illuminate the activity of reacting to other Facebook users' posts by drawing upon the theoretical framework recently developed to conceptualise the sense of commitment [1–4]. Using this framework, we hypothesized that, by performing actions on Facebook (e.g., posting or reacting to a post, or even just passively consuming), one may generate expectations about future actions one will perform on Facebook (what one will react to or post about, what one will read, how often one will be on Facebook, etc.), and consequently experience a sense of commitment to conform to these expectations. In particular, our question was whether agent A, through their behaviour on Facebook towards agent B (e.g., posting on B's Facebook wall or reacting to B's Facebook posts), A could generate a sense of being committed to B (e.g., to continuing these behaviours in the future), and also a sense that B is committed to A (e.g., that B is committed to posting on A's Facebook wall or to reacting to A's Facebook post as a result of A exhibiting these behaviours towards B). If that is right, this would explain how a particular kind of Facebook use—reacting to other users' posts—self-sustains over time, and moreover, it promises to shed light on some of the mechanisms which, when in overdrive, can lead to problematic Facebook use.

### Aims of the current study

Experiment 1 was designed to test whether participants would perceive a higher degree of commitment to react to a Facebook friend's post if they had done so regularly in the past than if this was not the case. In Experiment 2, we sought to generalise this by probing whether participants would feel more committed to react to a Facebook friend's post if they had regularly reacted to posts made by third parties. Experiment 3 inverted the perspective in the scenario from Experiment 1, probing whether participants would perceive a Facebook friend to be more committed to react to their post if that friend had reacted to similar posts in the recent past. Experiment 4 inverted the perspective in the scenario from Experiment 2, testing whether participants would perceive a Facebook friend to be committed to reacting to their post if that Facebook friend had regularly reacted to similar posts made by others.

## 2. Methods

### 2.1 Pre-registration and data availability

The hypotheses, sample size, methods, exclusion criteria and planned analyses for all four experiments were pre-registered before data collection, and can be accessed at: https:// aspredicted.org/blind.php?x=a4fg3w. The data and R code can be accessed here: https://osf.io/

vu6h4/?view_only=940e248296c44e9f92c2af357440b77c. All aspects of the study were carried out in accordance with the pre-registered protocol unless otherwise stated.

## 2.2 Participant recruitment

We used Prolific Academic to implement a web-based paradigm. Since each participant gave only one judgment per test question, and since online experiments produce greater variability than lab-based experiments, we expected a high variability in our dependent variables. We therefore opted for a large sample size (we aimed to recruit 100 participants per experiment). We included data from those participants who had already begun the respective experiment when Prolific registered that the target number had been reached. The number of participants per experiment and the number of excluded participants (if any) for each experiment are specified below.

In each experiment we implemented a between-subjects design, participants being randomly allocated either to the High Commitment condition or the Low Commitment condition. Depending on the condition to which they had been allocated, they received a vignette. The two vignettes varied across experiments, as detailed below.

## 2.3 Ethical approval

All participants gave their written informed consent prior to the respective experiment, and received a small monetary sum for their participation. The experiments were conducted in accordance with the Declaration of Helsinki and was approved by the Egyetesített Pszichológiai Kutatásetikai Bizottság (i.e., the United Ethical Review Board for Research in Psychology in Hungary), reference number 2016_053.

## 2.4 Experiment 1

No participants were excluded. The dataset included 126 participants (41 women, 84 men, 1 unspecified) between the ages of 18 and 59 ($M$ = 26.95 years, $SD$ = 8.94 years), all of whom were English-speaking adults.

Participants were randomly allocated either to the High Commitment condition ($N$ = 59) or the Low Commitment condition ($N$ = 67). They were then presented with one of the following vignettes, according to condition:

High Commitment condition: You and Sam are Facebook friends. Over the past six weeks, she has posted each week about a challenging situation at work. You reacted to her post every time. Today she posts something and you do not react to it.

Low Commitment condition: You and Sam are Facebook friends. Over the past six weeks, she has posted each week about a challenging situation at work. You reacted to her post once or twice. Today she posts something and you do not react to it.

Participants were then presented with the following test questions, always in this order:

- *Feeling Question*: "Do you feel at all badly about not commenting on her post today?" The rationale for this question was to implement an implicit measure of the sense of commitment from the perspective of the individual potentially disappointing another individual's expectation.

- *Disappointment Question*: "Do you think Sam will be at all disappointed that you did not comment today?" The rationale for this question was to implement an implicit measure of the sense of commitment from the perspective of the individual whose expectation may potentially have been violated.

- *Anticipation Question*: "Have you ever decided not to react to a friend's post in order to avoid feeling that you have to react to similar posts in the future?" The rationale for this

question was to implement a more direct test of whether participants are aware of the possibility that activity on Facebook can raise others' expectations about future activity on Facebook, and whether they may avoid such activity for this reason.

- *Attention Check Question*: "What is the capital of France?"

## 2.5 Experiment 2

After excluding one participant who failed to answer all test questions, the dataset included 122 participants (40 women, 80 men, 2 unspecified) between the ages of 18 and 48 ($M = 24.98$ years, $SD = 6.99$ years), all of whom were English-speaking adults.

Participants were randomly allocated either to the High Commitment condition ($N = 55$) or the Low Commitment condition ($N = 67$). They were then presented with one of the following vignettes, according to condition:

High Commitment: You and Kris and Sam are all Facebook friends and are in a Facebook group together. Last week Kris posted something in the group and you commented on it. Today, Sam posts something in the same group and you do not comment on it.

Low Commitment: You and Kris and Sam are all Facebook friends and are in a Facebook group together. Last week Kris posted something in the group and you did not comment on it. Today, Sam posts something in the same group and you do not comment on it.

The test questions were exactly the same as in Experiment 1.

## 2.6 Experiment 3

There were no excluded participants. The dataset therefore included 122 participants (48 women, 72 men, 2 unspecified) between the ages of 18 and 68 ($M = 26.64$ years, $SD = 8.79$ years), all of whom were English-speaking adults.

Participants were randomly allocated either to the High Commitment condition ($N = 66$) or the Low Commitment condition ($N = 56$). They were then presented with one of the following vignettes, according to condition:

High Commitment condition: You and Sam are Facebook friends. Over the past six weeks, you have posted each week about a challenging situation at work. Sam reacted to your post every time. Today you post something and she does not react to it.

Low Commitment condition: You and Sam are Facebook friends. Over the past six weeks, you have posted each week about a challenging situation at work. Sam reacted to your post once or twice. Today you post something and she does not react to it.

The wording of the feeling question and the disappointment question, as well as their order of presentation, were revised to reflect the shift in perspective:

- *Disappointment Question*: Do you feel at all disappointed that Sam did not react to your post today?

- *Feeling Question*: If you were Sam, do you think you would feel at all badly if you noticed the post but did not react to it?

The anticipation question and the attention check question were exactly the same as in experiments 1 and 2.

## 2.7 Experiment 4

After excluding ten participants who failed to answer all test questions, the dataset included 115 participants (53 women, 61 men, 1 unspecified) between the ages of 18 and 68 ($M = 26.93$ years, $SD = 8.77$ years), all of whom were English-speaking adults.

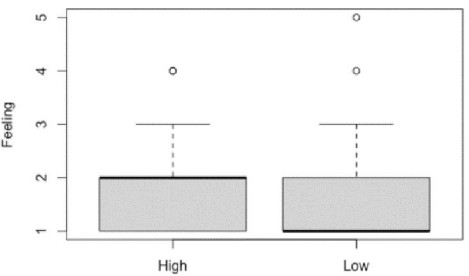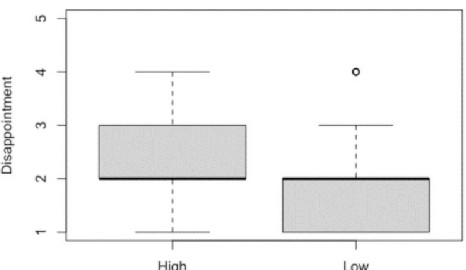

**Fig 1. Box and whisker plots displaying the median and interquartile range (IQR) of responses to the feeling and disappointment question in the two conditions.**

Participants were randomly allocated either to the High Commitment condition ($N = 61$) or the Low Commitment condition ($N = 54$). They were then presented with one of the following vignettes, according to condition:

High Commitment: You and Sam are Facebook friends. You have noticed that, over the last month, Sam has posted nearly every day on her Facebook wall or reacted to someone else's post. Yesterday, you wrote a post on your Facebook wall about a challenging situation at work. Sam has not reacted to it.

Low Commitment: You and Sam are Facebook friends. You have noticed that, over the last month, Sam has almost never posted on her Facebook wall or reacted to anyone else's post. Yesterday, you wrote a post on your Facebook wall about a challenging situation at work. Sam has not reacted to it.

The test questions were exactly the same as in Experiment 3.

## 3. Results

### 3.1 Experiment 1

We first confirmed that all participants answered all test questions, and that no participants failed to respond correctly (i.e. "Paris") to the attention check question.

For the Feeling Question (Do you feel at all badly about not commenting on her post today?), we first performed a Shapiro Wilk test, which revealed that the data were not distributed normally, $W = 0.71$, $p < .001$. Next, a Mann-Whitney U test showed that there was a significant difference between the High Commitment condition ($Mdn = 2$) and the Low Commitment Condition ($Mdn = 1$), $W = 2377$, $p = .026$, $r = .199$ (See **Fig 1**).

For the Disappointment Question (Do you think Sam will be at all disappointed that you did not comment today?), we first performed a Shapiro Wilk test, which revealed that the data were not distributed normally, $W = 0.85$, $p < .001$. Then, a Mann-Whitney U test showed that there was a significant difference between the High Commitment condition ($Mdn = 2$) and the Low Commitment Condition ($Mdn = 2$), $W = 2454.5$, $p = .013$, $r = .221$ (See **Fig 1** and **Table 1**).

**Table 1. Overview of results.** The table provides an overview of the results from Experiments 1–4. Values in bold indicate a statistically significant results, assuming an alpha level of .05.

| Exp | Feeling | Disappointment | Correlation Anticipation-Feeling | Correlation Anticipation-Disappointment | Correlation Feeling-Disappointment |
|---|---|---|---|---|---|
| 1 | **p = .026** | **p < .001** | p = .855 | p = .050 | **p < .001** |
| 2 | **p = .004** | **p < .001** | p = .312 | p = .051 | **p < .001** |
| 3 | p = .429 | p = .356 | **p = .007** | p = .596 | **p < .001** |
| 4 | **p = .017** | **p = .020** | **p = .002** | **p = .007** | **p < .001** |

Additionally, we conducted a Spearman correlation to probe the relationship between responses to the feeling question (Do you feel at all badly about not commenting on her post today?)and responses to the disappointment question (Do you feel at all disappointed that Sam did not react to your post today?). The results indicated that there was a statistically significant correlation between responses to the feeling question and responses to the disappointment question ($r_s = 0.425$, $p < .001$, $N = 126$).

Next, we conducted a Spearman correlation to test whether responses to the anticipation question (Have you ever decided not to react to a friend's post in order to avoid feeling that you have to react to similar posts in the future?) would predict responses to the feeling question (Do you feel at all badly about not commenting on her post today?). We predicted that participants giving high responses to the anticipation question would also report feeling worse in the imagined scenario specifically in the High Commitment condition. For this analysis, we included only those participants in the High Commitment condition. The results indicated that there was no statistically significant correlation between responses to the anticipation question and responses to the feeling question ($r_s = 0.024$, $p = .855$, $N = 59$).

Finally, we performed a Spearman correlation to test whether responses to the anticipation question (Have you ever decided not to react to a friend's post in order to avoid feeling that you have to react to similar posts in the future?) would predict responses to the disappointment question (Do you feel at all disappointed that Sam did not react to your post today?). We predicted that participants giving high responses to the anticipation question would also attribute greater disappointment specifically in the High Commitment condition. Again, we included only those participants in the High Commitment condition. The results indicated that the correlation between responses to the anticipation question and responses to the disappointment question was marginally statistically significant ($r_s = 0.26$, $p = .050$, $N = 59$).

## 3.2 Experiment 2

We first excluded one participant who did not answer all test questions, and confirmed that no participants failed to respond correctly (i.e. "Paris") to the attention check question. Thus, one participant in total was excluded from the analyses.

For the Feeling Question (Do you feel at all badly about not commenting on her post today?), we first performed a Shapiro Wilk test, which revealed that the data were not distributed normally, $W = 0.73$, $p < .001$. Next, Mann-Whitney U test showed that there was a significant difference between the High Commitment condition ($Mdn = 2$) and the Low Commitment Condition ($Mdn = 1$), $W = 2347$, $p = .004$, $r = .263$ (See **Fig 2**).

For the Disappointment Question (Do you think Sam will be at all disappointed that you did not comment today?), we first performed a Shapiro Wilk test, which revealed that the data

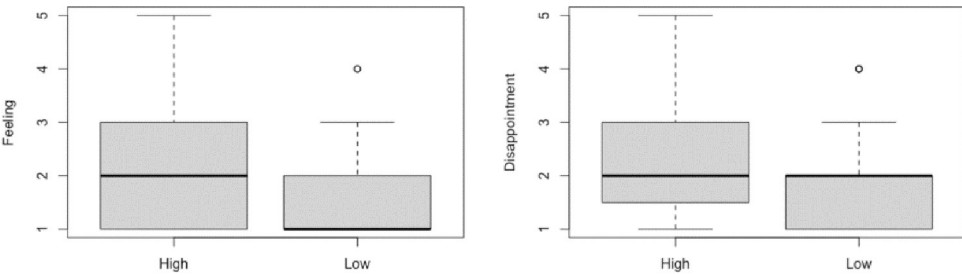

**Fig 2. Box and whisker plots displaying the median and IQR of responses to the feeling and disappointment question in the two conditions.**

were not distributed normally, $W = 0.83$, $p < .001$. Then, a Mann-Whitney U test showed that there was a significant difference between the High Commitment condition ($Mdn = 2$) and the Low Commitment Condition ($Mdn = 2$), $W = 2258.5$, $p = .022$, $r = .207$ (See **Fig 2** and **Table 1**).

Additionally, we conducted a Spearman correlation to probe the relationship between responses to the feeling question (Do you feel at all badly about not commenting on her post today?) and responses to the disappointment question. The results indicated that there was a statistically significant correlation between responses to the feeling question and responses to the disappointment question ($r_s = 0.588$, $p < .001$, $N = 122$).

Next, we conducted a Spearman correlation to test whether responses to the anticipation question (Have you ever decided not to react to a friend's post in order to avoid feeling that you have to react to similar posts in the future?) would predict responses to the feeling question (Do you feel at all badly about not commenting on her post today?). We predicted that participants giving high responses to the anticipation question would also report feeling worse in the imagined scenario specifically in the High Commitment condition. For this analysis, we included only those participants in the High Commitment condition. The results indicated that there was no statistically significant correlation between responses to the anticipation question and responses to the feeling question ($r_s = 0.139$, $p = .312$, $N = 55$).

Finally, we performed a Spearman correlation to test whether responses to the anticipation question (Have you ever decided not to react to a friend's post in order to avoid feeling that you have to react to similar posts in the future?) would predict responses to the disappointment question (Do you feel at all disappointed that Sam did not react to your post today?). We predicted that participants giving high responses to the anticipation question would also attribute greater disappointment specifically in the High Commitment condition. Again, we included only those participants in the High Commitment condition. The results indicated that the correlation between responses to the anticipation question and responses to the disappointment question was not statistically significant, but was approaching significance ($r_s = 0.091$, $p = .051$, $N = 55$).

### 3.3 Experiment 3

We first confirmed that all participants answered all test questions, and that no participants failed to respond correctly (i.e. "Paris") to the attention check question. Thus, no participants were excluded from the analyses.

For the Feeling Question (If you were Sam, do you think you would feel at all badly if you noticed the post but did not react to it?), we first performed a Shapiro Wilk test, which revealed that the data were not distributed normally, $W = 0.84$, $p < .001$. Next, a Mann-Whitney U test showed that there was no significant difference between the High Commitment condition ($Mdn = 2$) and the Low Commitment Condition ($Mdn = 2$), $W = 1703.5$, $p = .429$, $r = .072$ (See **Fig 3**).

For the Disappointment Question (Do you feel at all disappointed that Sam did not react to your post today?), we first performed a Shapiro Wilk test, which revealed that the data were not distributed normally, $W = 0.84$, $p < .001$. Then, a Mann-Whitney U test showed that there was no significant difference between the High Commitment condition ($Mdn = 2$) and the Low Commitment Condition ($Mdn = 2$), $W = 2019$, $p = .356$, $r = .084$ 072 (See **Fig 3** and **Table 1**).

Additionally, we conducted a Spearman correlation to probe the relationship between responses to the feeling question (Do you feel at all badly about not commenting on her post today?) and responses to the disappointment question (Do you feel at all disappointed that

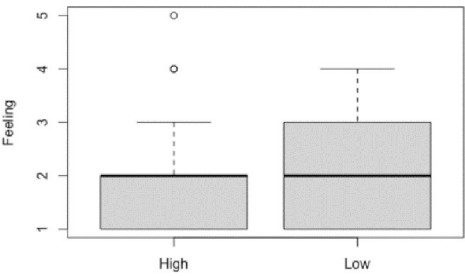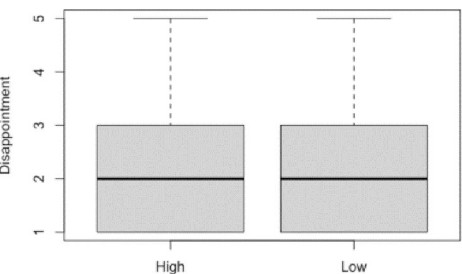

**Fig 3. Box and whisker plots displaying the median and IQR of responses to the feeling and disappointment question in the two conditions.**

Sam did not react to your post today?). The results indicated that there was a statistically significant correlation between responses to the feeling question and responses to the disappointment question ($r_s$ = 0.499, $p <$ .001, $N$ = 122).

Next, we conducted a Spearman correlation to test whether responses to the anticipation question (Have you ever decided not to react to a friend's post in order to avoid feeling that you have to react to similar posts in the future?) would predict responses to the feeling question (Do you feel at all badly about not commenting on her post today?). We predicted that participants giving high responses to the anticipation question would also expect the other party to feel badly in the imagined scenario specifically in the High Commitment condition. For this analysis, we included only those participants in the High Commitment condition. The results indicated that there was a statistically significant correlation between responses to the anticipation question and responses to the feeling question ($r_s$ = 0.331, $p$ = .007, $N$ = 66).

Finally, we performed a Spearman correlation to test whether responses to the anticipation question (Have you ever decided not to react to a friend's post in order to avoid feeling that you have to react to similar posts in the future?) would predict responses to the disappointment question (Do you feel at all disappointed that Sam did not react to your post today?). We predicted that participants giving high responses to the anticipation question would also report feeling more disappointed in the imagined scenario specifically in the High Commitment condition. Again we included only those participants in the High Commitment condition. The results indicated that the correlation between responses to the anticipation question and responses to the disappointment question was not statistically significant ($r_s$ = 0.066, $p$ = .596, $N$ = 66).

### 3.4 Experiment 4

We first excluded ten participants who did not answer all test questions, and confirmed that no participants failed to respond correctly (i.e. "Paris") to the attention check question. Thus, ten participants in total were excluded from the analyses.

For the Feeling Question (If you were Sam, do you think you would feel at all badly if you noticed the post but did not react to it?), we first performed a Shapiro Wilk test, which revealed that the data were not distributed normally, $W$ = 0.81, $p <$ .001. Next, a Mann-Whitney U test showed that there was a significant difference between the High Commitment condition ($Mdn$ = 2) and the Low Commitment Condition ($Mdn$ = 1), $W$ = 2050, $p$ = .017, $r$ = .224 (See **Fig 4**).

For the Disappointment Question (Do you feel at all disappointed that Sam did not react to your post today?), we first performed a Shapiro Wilk test, which revealed that the data were not distributed normally, $W$ = 0.78, $p <$ .001. Then, a Mann-Whitney U test showed that there

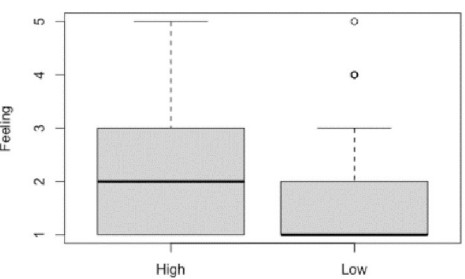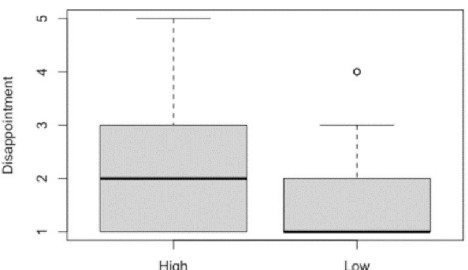

**Fig 4. Box and whisker plots displaying the median and IQR of responses to the feeling and disappointment question in the two conditions.**

was a significant difference between the High Commitment condition ($Mdn = 2$) and the Low Commitment Condition ($Mdn = 1$), $W = 2028$, $p = .020$, $r = .216$ (See **Fig 4** and **Table 1**).

Additionally, we conducted a Spearman correlation to probe the relationship between responses to the feeling question (Do you feel at all badly about not commenting on her post today?) and responses to the disappointment question (Do you feel at all disappointed that Sam did not react to your post today?). The results indicated that there was a statistically significant correlation between responses to the feeling question and responses to the disappointment question ($r_s = 0.588$, $p < .001$, $N = 115$).

Next, we conducted a Spearman correlation to test whether responses to the anticipation question (Have you ever decided not to react to a friend's post in order to avoid feeling that you have to react to similar posts in the future?) would predict responses to the feeling question (Do you feel at all badly about not commenting on her post today?). We predicted that participants giving high responses to the anticipation question would also expect the other party to feel badly in the imagined scenario specifically in the High Commitment condition. For this analysis, we included only those participants in the High Commitment condition. The results indicated that there was a statistically significant correlation between responses to the anticipation question and responses to the feeling question ($r_s = 0.139$, $p = .002$, $N = 61$).

Finally, we performed a Spearman correlation to test whether responses to the anticipation question (Have you ever decided not to react to a friend's post in order to avoid feeling that you have to react to similar posts in the future?) would predict responses to the disappointment question (Do you feel at all disappointed that Sam did not react to your post today?). We predicted that participants giving high responses to the anticipation question would also report feeling more disappointed in the imagined scenario specifically in the High Commitment condition. Again we included only those participants in the High Commitment condition. The results indicated that the correlation between responses to the anticipation question and responses to the disappointment question was statistically significant ($r_s = 0.34$, $p = .007$, $N = 61$).

## 4. Discussion

The results from Experiments 1, 2 and 4 (though not Experiment 3) provided evidence that regularly commenting on others' Facebook posts generated a sense of commitment to commenting on similar posts in the future, leading people to feel worse about not commenting on a post if they have done so regularly in the past than if they have not done so, and leading them to expect a Facebook friend to be more disappointed if they do not comment when there has been this previous history of commenting. These conclusions are corroborated by the finding that responses to the Anticipation question significantly predicted responses to the Feeling

question as well as the Disappointment question in the High Commitment condition in Experiment 4, and almost significantly predicted responses to the Disappointment question in the High Commitment condition in Experiments 1 and 2—these findings suggest that those participants who were more prone to experiencing a sense of commitment in Facebook interactions also experienced more negative emotions in response to the test questions in the High Commitment condition. It must be noted, however, that the correlation between responses to the Anticipation question and responses to the Feeling question in the High Commitment condition did not reach significance in Experiments 1 and 2.

This builds upon research showing that repeatedly participating in a joint action can give rise to a sense of commitment to continue one's participation [1–4]. The current results extend this earlier research by showing that this is also the case for online interactions on social media, a more indirect form of interaction than previously investigated.

The results from Experiments 2 and 4 also extend previous research in one additional respect. Previous studies have investigated interactions between two agents: A's behaviour towards B generates a sense that A is committed towards B. Experiments 2 and 4 extend this by showing that A's behaviour towards third parties can generate a sense of commitment to B; in Experiment 2, this sense of commitment is generated by A's behaviour towards C, and in Experiment 4 it is generated by A's behaviour towards a multiplicity of individuals. Presumably, it is crucial in this respect that the agent making the judgment is aware of the other agent's behaviour towards those third parties. For this reason, the vignettes in Experiment 2 were constructed such that the two agents were part of the same Facebook group, since members of the same group would likely be aware of each other's activity. Moreover, the vignette in Experiment 4 was clearly formulated in such a way that the agent making the judgment was aware of Sam's behaviour towards Facebook users other than herself or himself.

It is noteworthy that Experiment 3 did not yield significant results. One possible reason for this is that the vignettes were narrated from the perspective of the agent making the judgment; the agent is asked both how disappointed they are that Sam did not reply to their post and how bad they (the participant) feel about this. We speculate that, when taking their own point of view, the participant is aware that they are not particularly disappointed about Sam's failure to answer to their post, which explains their response to the Disappointment question. We conjecture that the participant's response to the Disappointment question then drives their response to the Feeling question: since the participant does not feel particularly disappointed by Sam's behaviour, they also do not imagine that Sam will feel especially bad about not posting.

This conjecture is buttressed by the finding that responses to the Feeling and Disappointment questions were significantly correlated in all four experiments. This raises the question whether the first question participants were asked may have primed their responses to subsequent questions. It is not possible to answer this question on the basis of the data collected here, since we maintained the same order of questions for all participants in each experiment (the Feeling question first in Experiments 1 and 2, the Disappointment question first in Experiments 3 and 4). The reason for this was that we wanted to prioritise the question directed to the experience of the participant in the imagined scenario—i.e. asking how they would feel (Experiments 1 and 2) or whether they would be disappointed (Experiments 3 and 4)—rather than the question about the other agent's presumed experience—i.e. whether the other agent would likely be disappointed (Experiments 1 and 2) or how they would likely feel (Experiments 3 and 4). Investigating this issue further would be a valuable avenue for future research.

It is also interesting to compare the findings from Experiment 1 and Experiment 3. The vignette in Experiment 1 was from the point of view of someone other than the participant, and here the questions as to whether Sam will feel bad that the subject has not reacted to her

post, as well as whether the participant would feel at all bad about not reacting to Sam's post yielded significantly different results in the High Commitment vs. the Low Commitment condition. By contrast, in Experiment 3, there is no significant difference in the responses to the Feeling and Disappointment questions between the High Commitment and the Low Commitment conditions. We speculate that, in the kinds of scenarios considered in the vignettes, participants may hold themselves to higher standards than they hold other people to—something that we would not have predicted. Our results suggest the following asymmetry in mutual expectations about the sense of commitment: participants feel more pressure to react to other agents' posts, especially if they have been doing so regularly, than they think that others (e.g., Sam) are under pressure to react to their own posts, when considering them in isolation (e.g., Sam's Facebook behaviour in relation to the participant only). However, as soon as it is made salient to participants that a Facebook friend of theirs—Sam—has been reacting to other individuals' posts, then they begin to expect that Sam will exhibit a certain kind of behaviour towards them, too.

Our research also complements existing research on "likes". As mentioned, our study investigated the activity of *giving*, rather than *receiving*, "likes", and what sustains this activity over time. As mentioned previously, a study revealed that "liking" images posted by other people (i.e., a single instance of giving social currency) activates areas of the brain that are associated with reward, and that "liking" other people's posts might *feel good* in the way that other forms of giving social currency do [12].

In addition, our study illuminates another aspect of the phenomenon of giving "likes" as exchanging social currency. Specifically, it addresses the question as to what happens when one gives social currency repeatedly over time. We found evidence that this generates a sense that one is thereby committed to giving more social currency in the future, and that one might disappoint others by failing to do so. So, while reacting to another's post might feel good in the context of a single instance, doing so repeatedly over time might lead to an unpleasant feeling, namely a felt *fear of disappointing* others. Relatedly, the need to *avoid feeling bad* in the future might drive some people to give "likes".

This fits in well with the sociometer theory of self-esteem, according to which self-esteem is "an internal monitor of the degree to which one is valued (and devalued) as a relational partner" [9, p. 2]. It has been pointed out in the past how people might engage in Facebook activity for the purposes of enhancing self-esteem: as mentioned, posting items that will receive many "likes" enhances one's self-esteem [10, 11], and, more generally, seeking attention and pursuing acknowledgement from others are maintained to be some of the main drivers of Facebook use [13, 14], although this is appears to be less true of people with a greater sense of purpose [15]. Building on this previous literature, our study illuminates another side of the relationship between Facebook use and self-esteem: if one reacts to others' Facebook posts repeatedly, the need to protect one's self-esteem (with the latter interpreted as a sociometer) may be activated, by inducing in that user the sense that they now need to sustain that kind of activity in the future so as to avoid disappointing their Facebook friends. In other words, disappointing Facebook friends is a threat to one's sense of being valued in relational terms by others.

By highlighting the way in which repeatedly reacting to others' posts induces a sense of commitment to doing so in the future in order to avoid disappointing others, the current research also contributes to the growing body of research on the problematic use of Social Networking Sites (SNSs) such as Facebook. Problematic Facebook use has many facets, which include the possibility that Facebook use can become addictive, as well as the possibility that Facebook use can decrease one's well-being [7].

Indeed, it is interesting to note that certain kinds of Facebook use have been discussed in terms of addiction [6, 16–25]. Though the aptness of the concept of *addiction* is a matter of

debate, insofar as this label is typically reserved for problematic use of substances [25–29], it is very telling that the very idea of SNSs addiction is being discussed at all, insofar as this indicates that many SNS users are driven to use these sites regardless of any disincentives. Our finding that activity on Facebook does generate a sense of commitment in Facebook users sheds light on an additional factor potentially influencing the problematic use of this SNS: in some participants, feeling a sense of commitment might lead to increased activity on Facebook, unless and until participants start refraining from Facebook activity in order to escape this sense of commitment, as responses to the Anticipation question seem to indicate. We note that our study focused on activity on Facebook, where reactions to posts are not anonymous. On the basis of our results, we may conjecture that, if users were given the option to react anonymously to others' posts, this might reduce their anticipation that their commenting activity might generate expectations about future activity in other users. Since Facebook currently does not allow anonymous posts in the contexts relevant to the vignettes used in the current study, our results do not bear directly upon this issue, but future research may use the current study as a starting point in exploring this.

Lastly, by highlighting the way in which repeated reaction to others' posts on Facebook generates the sense that others may be disappointed if this activity is not sustained in the future, our study contributes to another growing body of research that connects SNSs activity with lower levels of well-being [30, 31], anxiety and depression [32–36].

It is possible that the perceived pressure to increase activity on Facebook as a result of existing activity in order to avoid disappointing others and decreasing one's relational value in their eyes could generate negative feelings, e.g., anxiety. These feelings might be among the factors that contribute to decreasing well-being.

It must also be acknowledged that the current study also had several limitations. On the one hand, within Facebook activity, more distinctions could be drawn. For example, how big the Facebook group considered in Experiment 2 is, and whether this affects the perceived degree of sense of commitment: a smaller group might lead to an increased sense that all members of the group should be treated equally in terms of reactions to their posts. It would also be interesting to consider a wider variety of Facebook activity—whether for professional purposes, or for keeping in touch with friends, or in order to promote causes that are close to one's heart, since it has been shown that the detrimental emotional effects of Facebook are less marked in individuals who have a greater sense of meaning [8].

Another limitation is that the only SNS considered in the present research is Facebook. Future research should investigate other forms of SNS and social media, which may offer different possibilities for interaction than those found on Facebook—for example, SNSs such as Instagram, or else social media such as Whatsapp. In addition, it would be valuable to explore cross-cultural differences in people's experience of the sense of commitment in the context of interactions on Facebook and other forms of social media: the current study involved participants based in the UK, so we must be cautious about extrapolating the results to other cultural contexts. Relatedly, while the present study assumes that the identity of a user who reacts to a post is not hidden, it would be interesting to explore sense of commitment in social media that hide "likes". Our present results suggest that the reacting anonymously to a post would generate less pressure to continuing to react to that user's posts in the future. Finally, it may also be fruitful for future research to investigate to what extent people's experience of a sense of commitment in the context of SNS may be related to other constructs, such as self-esteem, mood, social anxiety, etc.

In sum, this research shows that commenting on others' Facebook posts generates a sense of commitment to further similar activity on social media, extending research on the sense of commitment to online interactions. The findings reported here present a novel perspective on

compulsive social media use and its effects on well-being, and open up new avenues for research on people's attitudes and behaviour in online interactions.

## Acknowledgments

We would like to thank the members of the Sense of Commitment Lab, who provided generous feedback in the context of lab meetings at various stages, and the audience of the ESPP 2021 conference in Leipzig.

## Author Contributions

**Conceptualization:** Chiara Brozzo, John Michael.

**Data curation:** John Michael.

**Formal analysis:** John Michael.

**Funding acquisition:** John Michael.

**Investigation:** Chiara Brozzo, John Michael.

**Methodology:** Chiara Brozzo, John Michael.

**Project administration:** Chiara Brozzo, John Michael.

**Resources:** John Michael.

**Software:** John Michael.

**Supervision:** Chiara Brozzo, John Michael.

**Validation:** John Michael.

**Visualization:** John Michael.

**Writing – original draft:** Chiara Brozzo, John Michael.

**Writing – review & editing:** Chiara Brozzo, John Michael.

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
