## [Decision Letter · Decision Letter 0]

16 May 2022

PONE-D-22-11480A Sense of Commitment to Activity on Facebook

PLOS ONE

Dear Dr. Michael,

Thank you for submitting your manuscript to PLOS ONE. After careful consideration, we feel that it has merit but does not fully meet PLOS ONE’s publication criteria as it currently stands. Therefore, we invite you to submit a revised version of the manuscript that addresses the points raised during the review process.

Based on the review comments received from both reviewers, the Academic Editor feels that your manuscript could be reconsidered for publication provided you are willing to incorporate major revisions.

While preparing your revised manuscript, you are advised to carefully consider the reviewers comments which are attached and submit a list of responses to the comments.

We look forward to receiving your revised manuscript.

Kind regards,

Rano Mal Piryani, MBBS, MCPS, DTCD, MD, Fellowship in Med Education

Academic Editor

PLOS ONE

Journal Requirements:

2. Please change "female” or "male" to "woman” or "man" as appropriate, when used as a noun (see for instance https://apastyle.apa.org/style-grammar-guidelines/bias-free-language/gender).

Reviewers' comments:

Reviewer's Responses to Questions

**Comments to the Author**

1. Is the manuscript technically sound, and do the data support the conclusions?

Reviewer #1: No

Reviewer #2: Yes

2. Has the statistical analysis been performed appropriately and rigorously? 

Reviewer #1: No

Reviewer #2: Yes

3. Have the authors made all data underlying the findings in their manuscript fully available?

Reviewer #1: Yes

Reviewer #2: Yes

4. Is the manuscript presented in an intelligible fashion and written in standard English?

Reviewer #1: No

Reviewer #2: Yes

5. Review Comments to the Author

Reviewer #1: Thank you for considering me to review this manuscript (“A Sense of Commitment to Activity on Facebook”). The authors have drawn evidence to show that ‘’face-to-face dyadic interactions create a sense of commitment even in the absence of explicit agreements”, then posed an empirical question on whether such a social phenomenon is likely to occur when the interaction is impersonal as is in the case of online interactions. The authors have reported that those participants who often react by regularly commenting on others’ Facebook posts are more likely to develop a sense of commitment, that is, to feel obliged to insert “likes” in the social media posts. The authors interpreted that such positive comments on social media could lead to some of the traps of overindulgence in social media. The effect of social media in particular among the young population is increasingly coming to prominence due to the spike in poor mental health outcomes and to lack of vicissitudes in life among children and adolescents. Although the topic is "hot", the conceptualization of the study and its methodology appears to lack scientific vigour. In my reading, PLOS is known as having a policy stating that “papers are not to be excluded based on lack of perceived importance or adherence to a scientific field”. For these reasons, I proceeded to examine this manuscript, and some of the points on which the authors reflect are detailed below.

TITLE

#1). The title ("A Sense of Commitment to Activity on Facebook") must be extended to alert the reader to the approaches that were used to solicit the reported result. Is this a quasi-experimental design? Web-based observational paradigm? Vignette study? Something to color the title will boost the reader's interest.

INTRODUCTION

#2). As the authors have written the introduction as something akin to ‘blog’, I would encourage them to re-write the introduction. Lately, there has been criticism that many social science and psychological articles are not based on a certain theoretical framework (Front Psychol. 2021 Jan 28;12:609802. doi: 10.3389/fpsyg.2021.609802.). First of all, recognizing the work of Anthony L would also be critical. Burrow from Cornell University, Ithaca, NY, United States. Related to this is the theory known as sociometer theory [(Leary & Baumeister. The nature and function of self-esteem: Sociometer theory. In M.P. Zanna (Ed.), Advances in experimental social psychology, Academic Press, San Diego, CA, US (2000), pp. 1-62)]. Secondly, some theoretical background could be drawn from the authors' work on the phenomenology of mental effort/cost-benefit analysis (e.g. Review of Philosophy and Psychology (2021) 12:889–904. https://doi.org/10.1007/s13164-020-00512-7). Thirdly, I could see the entry into social psychology (for example group behavior and social learning theory) as another potential theoretical background for this manuscript. Relevant operant and classical conditioning as well as the role of neural mechanisms could also be brought forth to increase scholarship of the narrative in the introduction [e.g. Sherman, L. E., Payton, A. A., Hernandez, L. M., Greenfield, P. M., & Dapretto, M. (2016). The power of the ‘like’ in adolescence: Effects of peer influence on neural and behavioral responses to social media. Psychological Science, 27(7), 1027–1035. doi.org/10.1177/0956797616645673].

#3). At the end of the Introduction, the authors could alert the reader to the aims of the present study. The aims should reflect the contents of what was later described as ‘Experiment 1’, ‘Experiment 2’, ‘Experiment 3’, and ‘Experiment 4’.

Also, there are conceptual issues for the authors to consider: Perhaps ‘COMMENTS’ would have a different intention than ‘likes’ as the former is more ‘active’ and can include opinion, question, praise, etc. May require further clarity in the introduction.

METHOD

#4). With the popularity of social media, it is not clear why the authors only accrue a relatively small sample size in a quantitative study. This has led to reliance on non-parametric and descriptive statistics which, in turn, has reduced the methodological robustness.

#5). The rationale for sticking to this particular age group (ages of 18 and 59) should be described. The majority of the literature suggests, though there is an exception, that social media is dominated by teens and young adults.

#6). It would be better if the Method and Results are separated. This would require a detailed and succinct method section. The different studies could be narrated with different aims and the accompanying vignettes (see comments #9 and #11).

#7). This sentence (“The experiment was conducted in accordance with the Declaration of Helsinki and was approved by the (EPKEB) United Ethical Review Board for Research in Psychology”) should have a subheading. Suggestion: “Ethical approval”.

#8). Please spell out ‘EPKEB’

#9). The outcome measures (“The test questions”) have cropped up in the write-up without the reader being given rationales for their inclusion. Did we do the internal validity of the test questions?

#10). Various studies have examined the moderator, predictors, or factors associated with positive social media feedback. Why are no other outcome measures included in the present study (e.g.. Self-esteem, mood, cognition, temperaments).

RESULTS

#11). As suggested above (comment #3), the result should have subheadings, say, for Experiment 1, Experiment 2, Experiment 3, and Experiment 2. Alternatively, these experiments should have been stated in terms of the aims. In the result section, there should be subheadings for each aim.

#12). As mentioned above, the method and the result could be separated.

DISCUSSION

#13). I believe that the authors have neglected many studies in the literature. The Discussion could be recapitulated within the existing research on “likes” on social media (See comments #2 and #14).

#14). The bulk of the cited papers has examined the relationship between social media usage and sleep, mental health problems, “addiction”, personality characteristics, cognitive status, self-esteem, subjective happiness self-perception, and academic motivation. While this literature points out the implication of the overuse of social media, the present study examines whether “likes” breed more “likes”. The literature should focus on the latter rather than the former.

#15). It is worthwhile to note that some social media outlets in Canada, Ireland, Italy, Japan, Brazil, Australia, and New Zealand have started to hide “likes”. This should be mentioned in the literature as a potential confounder that warrants a discussion as well as one of the potential limitations of the present study. This, in turn, would suggest that the participants were skewed toward the particular geographic area where “likes” are allowed.

#16). Studies are showing a significant difference when one has an option to comment ‘anonymously’ than otherwise. This could reduce ‘anticipation’. This may require comment in the manuscript.

REFERENCES

#17). The authors have employed 35 references. As detailed above, some of these could change and more relevant studies could be cited. Some of them are shown below.

Burrow, A. L., & Rainone, N. (2016). How many likes did I get?: Purpose moderates links between positive social media

feedback and self-esteem. Journal of Experimental Social Psychology, 69, 232-236,

doi.org/10.1016/j.jesp.2016.09.005.

Sherman, L. E., Payton, A. A., Hernandez, L. M., Greenfield, P. M., & Dapretto, M. (2016). The power of the like in

adolescence: Effects of peer influence on neural and behavioral responses to social media. Psychological Science, 27(7),

1027–1035. doi.org/10.1177/0956797616645673.

Tiggemann M, Hayden S, Brown Z, Veldhuis J. The effect of Instagram “likes” on women’s social comparison and body

dissatisfaction. Body image. 2018 Sep 1;26:90-7.

Sherman LE, Hernandez LM, Greenfield PM, Dapretto M. What the brain ‘Likes’: neural correlates of providing feedback

on social media. Social cognitive and affective neuroscience. 2018 Sep; 13(7): 699-707.

Reviewer #2: Dear Author,

The topic of the research is actual and interesting. The development of four different experiments allows to investigate thoroughly the topic.

Methods and results are rigorous and well-described. The statistical analysis is adeguate for the number of observations recorded.

6. PLOS authors have the option to publish the peer review history of their article (what does this mean?). If published, this will include your full peer review and any attached files.

Reviewer #1: No

Reviewer #2: No

---

## [Author Response · Author response to Decision Letter 0]

30 Jun 2022

PLOS ONE Response to Reviewers

Thank you for providing these thorough and constructive reviews, and for the opportunity to revise and resubmit our manuscript. We have made substantial changes in light of the suggestions made by the reviewers, and hope you will agree that the manuscript is now much stronger as a result. Our explanations of these changes are detailed below, interspersed with the reviewers’ comments, in italics and indicated by →. 

Reviewer #1: 

TITLE

#1). The title ("A Sense of Commitment to Activity on Facebook") must be extended to alert the reader to the approaches that were used to solicit the reported result. Is this a quasi-experimental design? Web-based observational paradigm? Vignette study? Something to color the title will boost the reader's interest.

→We have revised the title accordingly.

INTRODUCTION

#2). As the authors have written the introduction as something akin to ‘blog’, I would encourage them to re-write the introduction. Lately, there has been criticism that many social science and psychological articles are not based on a certain theoretical framework (Front Psychol. 2021 Jan 28;12:609802. doi: 10.3389/fpsyg.2021.609802.). First of all, recognizing the work of Anthony L would also be critical. Burrow from Cornell University, Ithaca, NY, United States. Related to this is the theory known as sociometer theory [(Leary & Baumeister. The nature and function of self-esteem: Sociometer theory. In M.P. Zanna (Ed.), Advances in experimental social psychology, Academic Press, San Diego, CA, US (2000), pp. 1-62)]. Secondly, some theoretical background could be drawn from the authors' work on the phenomenology of mental effort/cost-benefit analysis (e.g. Review of Philosophy and Psychology (2021) 12:889–904. https://doi.org/10.1007/s13164-020-00512-7). Thirdly, I could see the entry into social psychology (for example group behavior and social learning theory) as another potential theoretical background for this manuscript. Relevant operant and classical conditioning as well as the role of neural mechanisms could also be brought forth to increase scholarship of the narrative in the introduction [e.g. Sherman, L. E., Payton, A. A., Hernandez, L. M., Greenfield, P. M., & Dapretto, M. (2016). The power of the ‘like’ in adolescence: Effects of peer influence on neural and behavioral responses to social media. Psychological Science, 27(7), 1027–1035. doi.org/10.1177/0956797616645673].

→ Thanks for these suggestions. We have thoroughly revised the introduction and the discussion along the lines suggested, and now include a number of the references suggested by the reviewer.

#3). At the end of the Introduction, the authors could alert the reader to the aims of the present study. The aims should reflect the contents of what was later described as ‘Experiment 1’, ‘Experiment 2’, ‘Experiment 3’, and ‘Experiment 4’.

→We have implemented this suggestion.

Also, there are conceptual issues for the authors to consider: Perhaps ‘COMMENTS’ would have a different intention than ‘likes’ as the former is more ‘active’ and can include opinion, question, praise, etc. May require further clarity in the introduction.

→This was phrased neutrally (in terms of "reacting"), so that people wouldn't interpret it specifically as either liking or commenting, but we thought that either activity would be sufficient to generate a sense of commitment in the Facebook user. Further studies may well investigate the specific contributions of liking as opposed to commenting.

METHOD

#4). With the popularity of social media, it is not clear why the authors only accrue a relatively small sample size in a quantitative study. This has led to reliance on non-parametric and descriptive statistics which, in turn, has reduced the methodological robustness.

→The reason why we did not choose even larger sample sizes was that we expected to observe medium-sized effects. It is certainly true that the lack of a normal distribution worked against us insofar as non-parametric stats are less sensitive, but given that we found significant results for our test questions in experiments 1,2 and 4, we do not believe that the sample sizes were inappropriately small. It is also worth noting that the Burrow and Rainone 2016 and the Sagioglou and Greitemeyer 2014 studies also had around 100 participants each in some of their studies. Future researchers who wish to follow up on the current research can use our effect sizes as a basis for making informed decisions about sample sizes and planned analyses in similar studies.

#5). The rationale for sticking to this particular age group (ages of 18 and 59) should be described. The majority of the literature suggests, though there is an exception, that social media is dominated by teens and young adults.

→ There was no upper limit; the lower limit of 18 was dictated by our ethics committee. Facebook reports that their users are distributed fairly well across different age groups:

119.0 million users aged 13 to 17 (5.6% of Facebook’s total ad audience)

481.1 million users aged 18 to 24 (22.4% of Facebook’s total ad audience)

643.2 million users aged 25 to 34 (30.1% of Facebook’s total ad audience)

401.5 million users aged 35 to 44 (18.7% of Facebook’s total ad audience)

237.0 million users aged 45 to 54 (11.0% of Facebook’s total ad audience)

145.4 million users aged 55 to 64 (6.8% of Facebook’s total ad audience)

114.1 million users aged 65 and above (5.3% of Facebook’s total ad audience)

This data is from: https://datareportal.com/essential-facebook-stats

#6). It would be better if the Method and Results are separated. This would require a detailed and succinct method section. The different studies could be narrated with different aims and the accompanying vignettes (see comments #9 and #11).

→We have implemented this suggestion.

#7). This sentence (“The experiment was conducted in accordance with the Declaration of Helsinki and was approved by the (EPKEB) United Ethical Review Board for Research in Psychology”) should have a subheading. Suggestion: “Ethical approval”.

→We have implemented this suggestion.

#8). Please spell out ‘EPKEB’

→We have implemented this suggestion.

#9). The outcome measures (“The test questions”) have cropped up in the write-up without the reader being given rationales for their inclusion. Did we do the internal validity of the test questions?

→We now include rationales for each test question. In the results section, we report the results of regressions probing the relationships among responses to the different test questions, which target distinct but complementary attitudes.

#10). Various studies have examined the moderator, predictors, or factors associated with positive social media feedback. Why are no other outcome measures included in the present study (e.g.. Self-esteem, mood, cognition, temperaments).

→ Given that this was a hypothesis-driven study, we were careful not to collect any data that was not directly relevant to testing our hypotheses. We certainly agree that it would be very valuable to carry out future research investigating how people’s experience of a sense of commitment on Facebook and other forms of social media might relate to outcome measures like the ones flagged by the reviewer. We have added a note to this effect in the last sentence of the penultimate paragraph of the discussion section.

RESULTS

#11). As suggested above (comment #3), the result should have subheadings, say, for Experiment 1, Experiment 2, Experiment 3, and Experiment 2. Alternatively, these experiments should have been stated in terms of the aims. In the result section, there should be subheadings for each aim.

→We have implemented this suggestion.

#12). As mentioned above, the method and the result could be separated.

→We have implemented this suggestion.

DISCUSSION

#13). I believe that the authors have neglected many studies in the literature. The Discussion could be recapitulated within the existing research on “likes” on social media (See comments #2 and #14).

→We have added several paragraphs in the discussion linking the current research with this background literature.

#14). The bulk of the cited papers has examined the relationship between social media usage and sleep, mental health problems, “addiction”, personality characteristics, cognitive status, self-esteem, subjective happiness self-perception, and academic motivation. While this literature points out the implication of the overuse of social media, the present study examines whether “likes” breed more “likes”. The literature should focus on the latter rather than the former.

→We have added several paragraphs in the discussion linking the current research with this background literature.

#15). It is worthwhile to note that some social media outlets in Canada, Ireland, Italy, Japan, Brazil, Australia, and New Zealand have started to hide “likes”. This should be mentioned in the literature as a potential confounder that warrants a discussion as well as one of the potential limitations of the present study. This, in turn, would suggest that the participants were skewed toward the particular geographic area where “likes” are allowed.

→Yes, we now acknowledge this in the discussion.

#16). Studies are showing a significant difference when one has an option to comment ‘anonymously’ than otherwise. This could reduce ‘anticipation’. This may require comment in the manuscript.

→We now discuss this idea in the discussion. We agree that commenting anonymously might reduce "anticipation". In any case, Facebook doesn't yet allow anonymous posts in contexts like the ones features in our vignettes, so this doesn't affect the present study.

REFERENCES

#17). The authors have employed 35 references. As detailed above, some of these could change and more relevant studies could be cited. Some of them are shown below.

→We have implemented this suggestion, revising the introduction and discussion, as detailed above, to connect our study with relevant background research.

Reviewer 2:

I suggest to merge the paragraphs (with relative sub-para- graphs) named:

2.1 Method, 3.1 Method, 4.1 Method, 5.1 Method

I suggest to merge even the paragraphs named: 

2.2 Results, 3.2 Results, 4.2 Results, 5.2 Results 

I understand you decided to maintain the four experiments divided, but it is very confusing. I suggest to create a uni- que section named ‘Method’ with a description of each ex- periment in sub-paragraphs. Evenly I suggest to create a unique section ‘Results’ with the four descriptions of resul- ts divided in sub-paragraphs. 

→We have restructured the paper as suggested.

Please add a legend to the figures 

→We have included figure legends.

Please for Table 1 add some informations in the legend (for example the meaning of p value reported in parenthesis) 

→We have revised the table to make it clearer what information the table is supposed to convey – i.e. it includes all the p values, with the significant ones in bold, assuming an alpha level of .05.

---

## [Decision Letter · Decision Letter 1]

8 Jul 2022

A Sense of Commitment to Activity on Facebook:Evidence from a web-based paradigm

PONE-D-22-11480R1

Dear Dr. Michael,

We’re pleased to inform you that your manuscript has been judged scientifically suitable for publication and will be formally accepted for publication once it meets all outstanding technical requirements.

Kind regards,

Rano Mal Piryani, MBBS, MCPS, DTCD, MD, Fellowship in Med Education

Academic Editor

PLOS ONE

Additional Editor Comments (optional):

Reviewers' comments:

Reviewer's Responses to Questions

**Comments to the Author**

1. If the authors have adequately addressed your comments raised in a previous round of review and you feel that this manuscript is now acceptable for publication, you may indicate that here to bypass the “Comments to the Author” section, enter your conflict of interest statement in the “Confidential to Editor” section, and submit your "Accept" recommendation.

Reviewer #1: All comments have been addressed

Reviewer #2: All comments have been addressed

2. Is the manuscript technically sound, and do the data support the conclusions?

Reviewer #1: Yes

Reviewer #2: Yes

3. Has the statistical analysis been performed appropriately and rigorously? 

Reviewer #1: I Don't Know

Reviewer #2: Yes

4. Have the authors made all data underlying the findings in their manuscript fully available?

Reviewer #1: Yes

Reviewer #2: Yes

5. Is the manuscript presented in an intelligible fashion and written in standard English?

Reviewer #1: Yes

Reviewer #2: Yes

6. Review Comments to the Author

Reviewer #1: The esteemed authors have ‘labored’ extensively to accommodate and sometimes rebut my comments. The authors' vigilance paid off. The manuscript has significantly improved both in terms of the conceptual issue as well as the scientific merit. In this regard, I have no other comments. I feel the manuscript could be accepted for publication.

Reviewer #2: (No Response)

7. PLOS authors have the option to publish the peer review history of their article (what does this mean?). If published, this will include your full peer review and any attached files.

Reviewer #1: **Yes: **Samir Al-Adawi

Reviewer #2: No

---

## [Editor Report · Acceptance letter]

13 Jul 2022

PONE-D-22-11480R1 

A Sense of Commitment to Activity on Facebook: Evidence from a web-based paradigm 

Dear Dr. Michael:

I'm pleased to inform you that your manuscript has been deemed suitable for publication in PLOS ONE. Congratulations! Your manuscript is now with our production department. 

Kind regards, 

on behalf of

Dr. Rano Mal Piryani 

Academic Editor

PLOS ONE